# The Development of 170 GHz, 1 MW Gyrotron for Fusion Application

**Yichi Zhang** [1,*], **Xu Zeng** [1], **Ming Bai** [2], **Ming Jin** [3], **Wenteng Hao** [1], **Dongshuo Gao** [1], **Qiao Liu** [1] **and Jinjun Feng** [1]

1 National Key Laboratory of Science and Technology on Vacuum Electronics, Beijing Vacuum Electronics Research Institute, Beijing 100015, China; zengxu1108@163.com (X.Z.); hwt1997113@163.com (W.H.); gaodongshuo0217@163.com (D.G.); q.liu.ah@outlook.com (Q.L.); fengjinjun@tsinghua.org.cn (J.F.)
2 School of Electronic Information Engineering, Beijing University of Aeronautics and Astronautics, Beijing 100871, China; mbai@buaa.edu.cn
3 College of Information Science and Technology, Beijing University of Chemical Technology, Beijing 100129, China; jinming@mail.buct.edu.cn
* Correspondence: zhangyichi1228@163.com

**Abstract:** This paper presents the overall conceptual design of a 170 GHz, 1 MW gyrotron for plasma heating applications in thermonuclear fusion reactors. The operating mode is carefully selected with consideration of mode competition. The $TE_{25,10}$ mode is determined as the operating mode in the present study. A weakly tapered conventional resonator is used for the study of the RF behavior, and multimode calculations are carried out for power and efficiencies. The optimized structure of the beam tunnel can further attenuate low-frequency oscillation. The design studies of a triode-type magnetic injection gun, quasi-optical mode converter, and single-disc sapphire window are also described. In addition, novel cold test methods which can effectively validate assembles are designed. In 2021, a prototype gyrotron was fabricated to validate the electronic–optic system and RF system. The current results obtained support an output power of 210 kW and efficiency of 15.9% through an initial low-power experiment. On the basis of the optimized design, an industrial prototype gyrotron is under fabrication as the heating source for the thermonuclear experimental reactor.

**Keywords:** gyrotron; 170 GHz; $TE_{25,10}$ mode; fusion

## 1. Introduction

For fusion reactions, high temperatures are required in magnetically confined plasmas. Initially, neutral beam injection or RF wave heating can provide power, heating the plasma up to 107 K [1]. Electron cyclotron resonance heating (ECRH) was proven as an efficient approach for EAST (Experimental Advanced Superconducting Tokamak). As the major microwave source equipped in the ECRH and current drive system, a gyrotron with high output power, high efficiency, high frequency, and long pulse length is a main direction of study to satisfy the fusion application.

In the tokamak system, ECRH plans to employ twenty-four 1 MW, 170 GHz gyrotrons as the power source, which may provide 20 MW of power injected to the ITER plasma, assuming a transmission efficiency of 83% [2]. The EDA phase (engineering design activities) of a 170 Hz gyrotron was carried out by Europe, Russia and Japan for a long time [3–5]. Before the 2000s, the high-power long-pulse gyrotron underwent important breakthrough technology, such as improving the efficiency of mode converter [6,7], designing a depressed collector [8,9], and upgrading the window material to diamond [4,10], thus realizing the technology route of the 170 GHz, 1 MW CW gyrotron. The Beijing Vacuum Electronics Research Institute (BVERI) designed a 170 GHz gyrotron, aiming to satisfy the basic performance requirement of the EAST gyrotron. In 2021, the 170 GHz gyrotron was manufactured and demonstrated with short pulses.

Figure 1 shows the schematic diagram of the gyrotron and the power supply configuration at BVERI. The gyrotron consists of five parts: (1) a triode magnetron injection gun (MIG), which has the advantage of adjusting the electron beam parameters by controlling the anode voltage; (2) between the MIG and the resonator, a beam tunnel structure is used to suppress parasitic oscillation, which consists of several alternating copper rings and attenuating ceramic rings; (3) the resonator is a hollow cylindrical weakly tapered cavity, where the electron beam interacts with the millimeter-wave radiation; (4) the converter consisting of a Denisov launcher and three mirrors can be used to transform the $TE_{25,10}$ mode to Gaussian quasi-optical mode; (5) a 1.4 mm sapphire window is employed to radiate the RF beam.

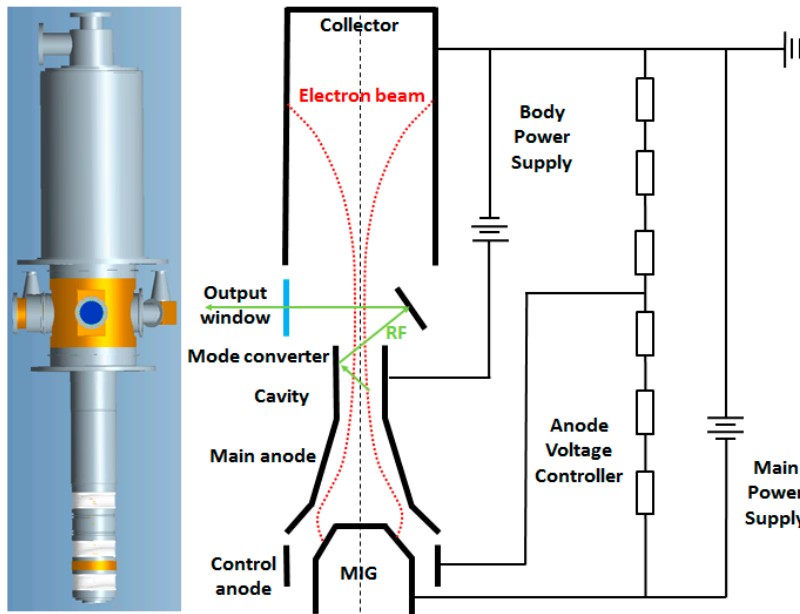

**Figure 1.** Model of 170 GHz gyrotron and power supply configuration.

By 2020, BVERI had completed the design and fabricated a $TE_{25,10}$ mode output gyrotron, aiming to validate the performance of the MIG, resonator, and sapphire window. Table 1 summarizes the design parameters of the 170 GHz gyrotron.

**Table 1.** The design parameters of the 170 GHz gyrotron.

| | |
|---|---|
| Operating mode | $TE_{25,10}$ |
| Frequency | 170 GHz |
| Accelerating voltage | 80 kV |
| Collector depression voltage | 27 kV |
| Beam current | 40 A |
| Cavity magnetic field | 6.72 T |
| Beam radius in the resonator $R_b$ | 7.4 mm |
| Average pitch angle | 1.32 |
| Cavity radius | 17.78 mm |
| RF output power | 1.06 MW |
| Electronic efficiency | 33.1% |
| Window material | Sapphire |

## 2. Mode Selection

In order to increase the power to 1 MW, the gyrotron employed a high-order volume mode to interact with electron beam. According to experience, eigenvalues ranging from 60 to 68 were considered for operating modes. The free space wavelength ($\lambda = c/f$) was

1.764 mm for the given frequency. For the operating modes TE$_{mn}$, the cavity radius is given by the following equation [11]:

$$R_0 = \frac{\chi_{m,n}\lambda}{2\pi} \approx 47.71 \frac{\chi_{m,n}}{f(GHz)}, \qquad (1)$$

where $\chi_{m,n}$ is the $n$-th root of $J'_m(\chi) = 0$. The radius of the 170 GHz cavity $R_0$ is about 17.8 mm. For the gyrotron operating at the first harmonic, the radius of electron beam in the resonator is related to the following equation [12]:

$$R_b = \frac{\chi_{m\pm1,1}R_0}{\chi_{m,n}}, \qquad (2)$$

where $\chi_{m\pm1,1}$ corresponds to the co-rotating and counter-rotating mode. For the quasi-optical output coupler with advanced dimpled wall launcher, the factor $k$ of candidate modes has to be considered, which is used to check the compatibility between mode and the launcher; the factor k is given by the following equation [12]:

$$k = \frac{\pi}{\arccos(m/\chi_{m,n})}. \qquad (3)$$

Equation (3) expresses the number of reflections per turn on the launcher, and the factor $k$ is usually about 3 for the ideal operating state. Table 2 lists a number of well-qualified modes according to the above equations.

**Table 2.** Characteristics of probable operating modes for 170 GHz gyrotron.

| m | n | $\chi_{m,n}$ | $R_0$ (mm) | $R_b$ (mm) | k |
|---|---|---|---|---|---|
| 21 | 11 | 61.5632 | 17.279 | 6.2362 | 2.6064 |
| 22 | 11 | 62.8612 | 17.643 | 6.5269 | 2.6263 |
| 24 | 10 | 62.0492 | 17.415 | 7.1073 | 2.7174 |
| 25 | 10 | 63.3197 | 17.772 | 7.3971 | 2.7374 |
| 27 | 9 | 62.3189 | 17.509 | 7.9760 | 2.8422 |
| 28 | 9 | 63.6259 | 17.858 | 8.2651 | 2.8624 |
| 30 | 8 | 62.5491 | 17.555 | 8.8428 | 2.9856 |
| 31 | 8 | 63.7675 | 17.897 | 9.1313 | 3.0060 |

To study the feasibility of the TE$_{25,10}$ mode operation, the starting currents under possible competing modes were calculated following a linearized single-mode theory [13,14], to determine whether the cavity was operating in the desired mode. Competing modes had closed starting currents distributed in different magnetic field values, as shown in Figure 2. The TE$_{25,10}$ mode could oscillate in a narrow range stably; Figure 3 shows that the nominalized coupling efficient value as a function of frequency while taking the co-rotation TE$_{25,10}$ as the main mode (the radius of electron guide center was 7.4 mm. From these two figures, we can find that the TE$_{22,11}$ and TE$_{23,11}$ modes have a similar magnetic field for the starting current and a similar frequency in terms of the coupling coefficient to the TE$_{25,10}$ mode than with other modes. The TE$_{22,11}$ and TE$_{23,11}$ modes have similar value of starting current and may avoid competition by accurately modulating the magnetic field. In fact, due to the coupling coefficient of co-rotation for the TE$_{22,11}$ and TE$_{23,11}$ modes being as low as possible, as shown in Figure 3, the competition of these two modes with TE$_{25,10}$ mode can be ignored. And in Figure 3, we can see that the major competing modes, TE$_{24,10}$ and TE$_{26,10}$ for TE$_{25,10}$, are fairly well separated. Considering the higher longitudinal index modes, ones can ignore the modes of TE$_{25,10,2}$ and TE$_{25,10,3}$, because they have much lower quality factors than the modes whose value of q is 1 (i.e., the TE$_{25,10,1}$ mode in the present case) [15,16]. Hence, the TE$_{25,10}$ mode was chosen as the operating mode for the 170 GHz gyrotron.

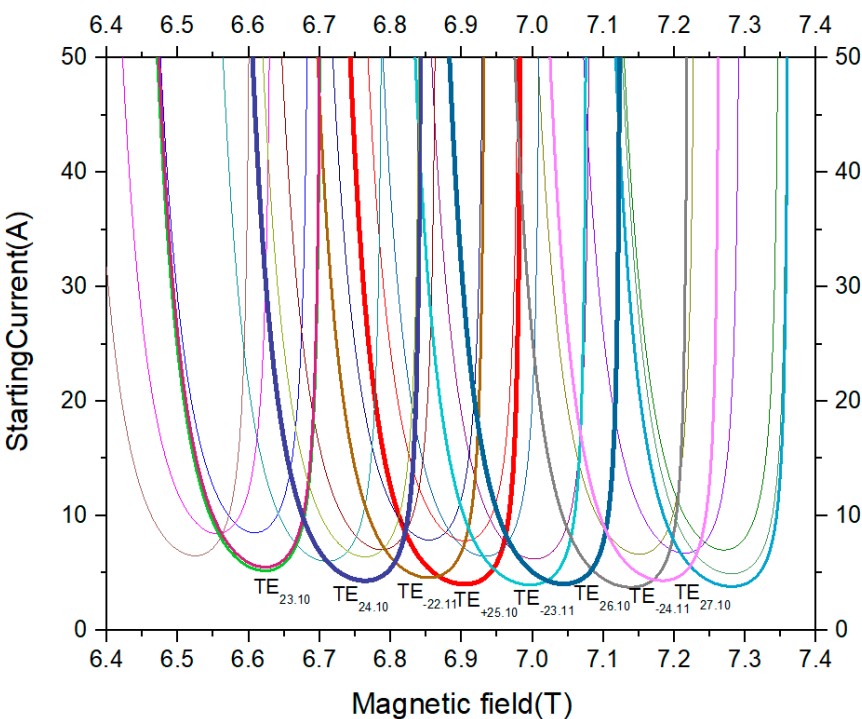

**Figure 2.** Starting current versus magnetic field for various modes operated at the nominal design parameters of the 170 GHz gyrotron.

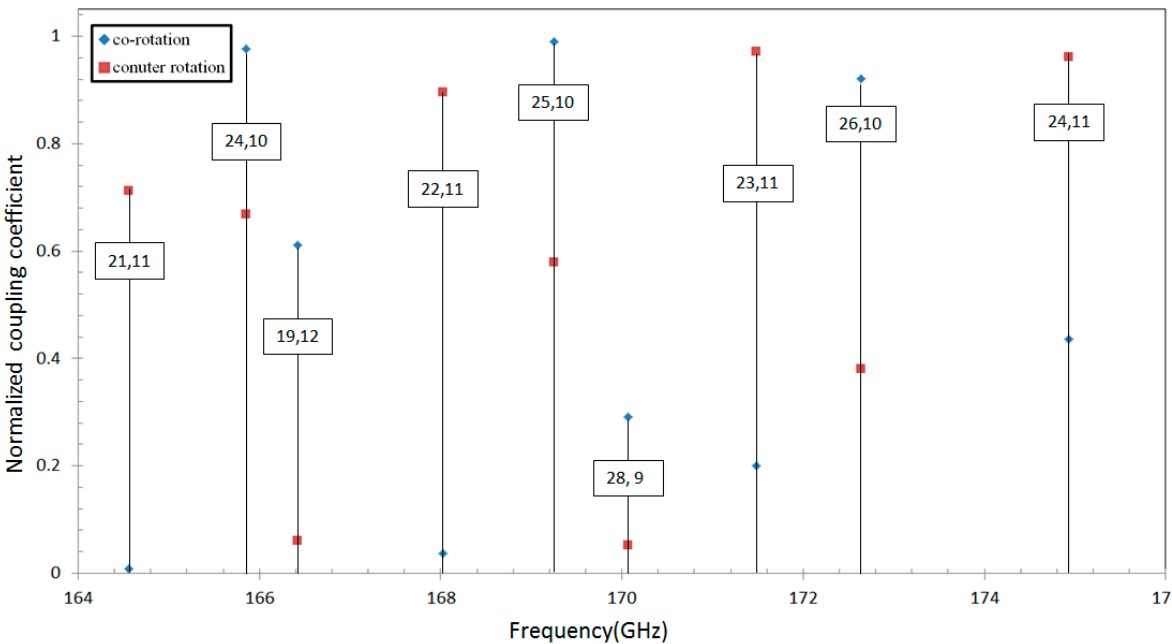

**Figure 3.** Normalized coupling efficiency for the competing modes at various frequency points. The TE$_{25,10}$ mode was chosen to the main mode for the 170 GHz gyrotron.

## 3. Design of 170 GHz Gyrotron

### *3.1. Cavity*

On the basis of the TE$_{25,10}$ mode, a hollow resonator with a down taper of 5° and an up taper of 3° was designed, as shown in Figure 4. The length of the cylindrical part was smoothly rounded over a length of 1.75 mm and 1.05 mm for input and output, which could prevent mode conversion in the cavity. According to a cold cavity simulation, the diffraction Q-value of the resonator was 1843.5 for the 170 GHz gyrotron.

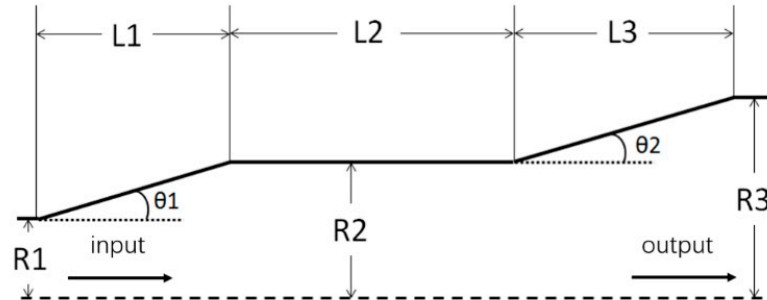

**Figure 4.** Schematic of the resonator for the 170 GHz gyrotron.

Figure 5 shows the output power versus magnetic field for the TE$_{25,10}$ mode with a beam current of 40 A and an accelerating voltage of 80 kV, corresponding to CW operation, guiding the choice of the best operating parameters for the experiments. It can be seen that 6.72 T was the optimized magnetic field for the 170 GHz resonator with the parameters shown in Table 3, with a peak power of about 1.1 MW and an efficiency of 35.3%. The simulation showed that the TE$_{24,10}$ mode would oscillate under the lower magnetic field and the peak power was less than that of the TE$_{25,10}$ mode.

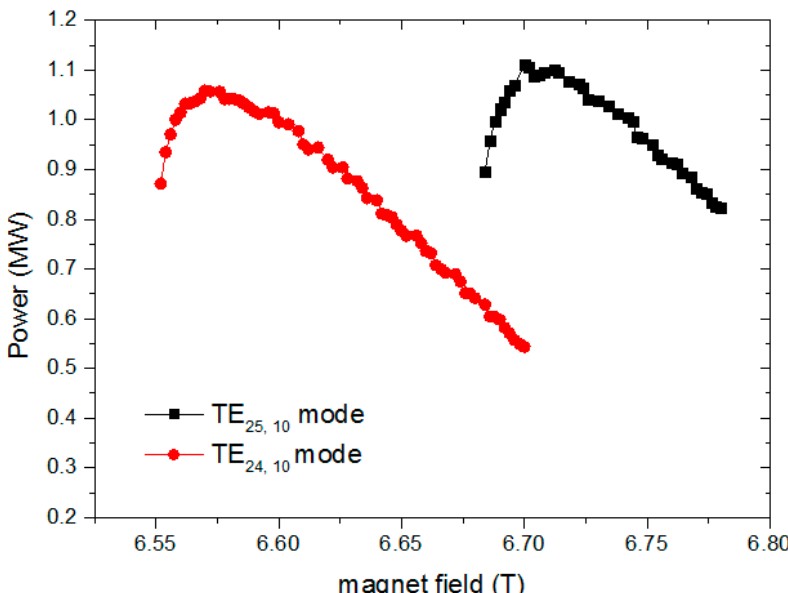

**Figure 5.** Simulation result of the gyrotron with cavity parameters listed in Table 3. The TE$_{25,10}$ mode and TE$_{24,10}$ mode oscillate in the nearby magnetic field.

**Table 3.** Design parameters for resonator of 170 GHz gyrotron.

| | |
|---|---|
| Frequency | 169.25 GHz |
| Cavity mode | TE$_{25,10}$ |
| $L_2$ | 15 mm |
| $R_2$ | 17.85 mm |
| $\theta_1$ | 5° |
| $\theta_2$ | 3° |
| Magnetic | 6.72 T |
| Beam radius $R_b$ | 7.4 mm |
| Voltage $U_c$ | 80 kV |
| Current $I$ | 40 A |
| Peak power | 1.06 MW |
| efficiency | 33.1% |

For the 1 MW gyrotron, the beam current and output power are large, which can exceed the starting current of other modes and cause parasitic oscillations. In [12], the authors focused on the beam tunnel; by measuring the cooling circuit temperature, the loss power was estimated as $1-5$ kW for 1 MW output power. During operation of the gyrotron, in order to maintain steady oscillation of the $TE_{25, 10}$ mode in every pulse length, it is necessary to modify the configuration of the beam tunnel to decrease the low-frequency oscillation. To further attenuate possible RF fields, a new beam tunnel structure was used with reference to the design of KIT, consisting of several copper rings and attenuating ceramic rings (BeO), instead of FeSiAl, which was previously coated on the beam tunnel surface. The optimized structure is shown in Figure 6.

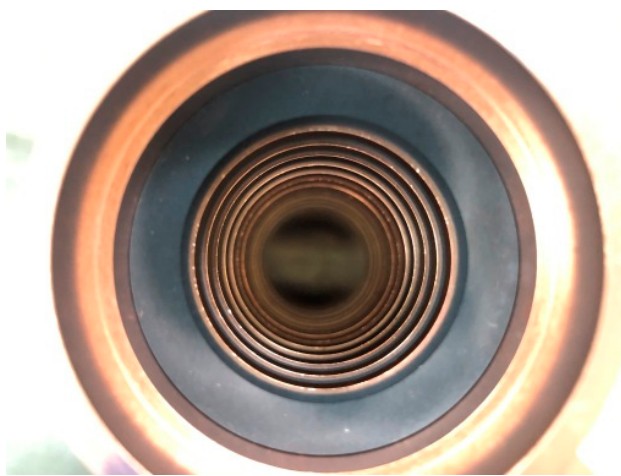

**Figure 6.** Image of the optimized beam tunnel structure.

### 3.2. Magnetron Injection Gun (MIG)

The MIG is the electron source in the gyrotron, consisting of an annular emitter mounted on a cathode, an accelerating anode, and a modulating anode for the triode-type case. It allows greater flexibility in controlling the beam parameters compared to the diode-type MIG configuration. The pitch factor $\alpha$ can be modulated independently with other parameters, which is the most significant advantage of the triode-type MIG. This means that the pitch factor $\alpha$, the electron cyclotron angular frequency in the cavity $\omega_c$, and the electron beam position in the cavity can be optimized independently by controlling the magnetic field in the resonator $B_c$ and modulating anode voltage $U_m$ during the oscillation.

A triode-type MIG has 19 independent geometric parameters. In order to achieve the target, according to the cavity calculation, the magnetic field strength compression ratio is 22, the distance between emitter rings and the cavity is 390 mm, and the magnetic field strength is approximately 0.29 T. The density of emitter $J_c$ is 3 A/cm$^2$ considering the cathode life with an operating current of 40 A. The numerical calculations were conducted using 2D code and CST trajectory studio with similar results. The electron trajectories of the optimized MIG are shown in Figure 7. The typically obtained beam properties were fairly good with the voltage of the cathode $U_c = 80$ kV, the voltage of the anode $U_a = 30$ kV, $\alpha = 1.3$, the spread of the transverse electron velocity $\Delta\beta_\perp = 2.49\%$, and the spread of the longitudinal electron velocity $\Delta\beta_\parallel = 4.52\%$. It was difficult to reduce the spread of velocity, due to the uniform distribution of magnetic field intensity on the cathode surface, as well as the strong space charge effect with the operating current of 40 A.

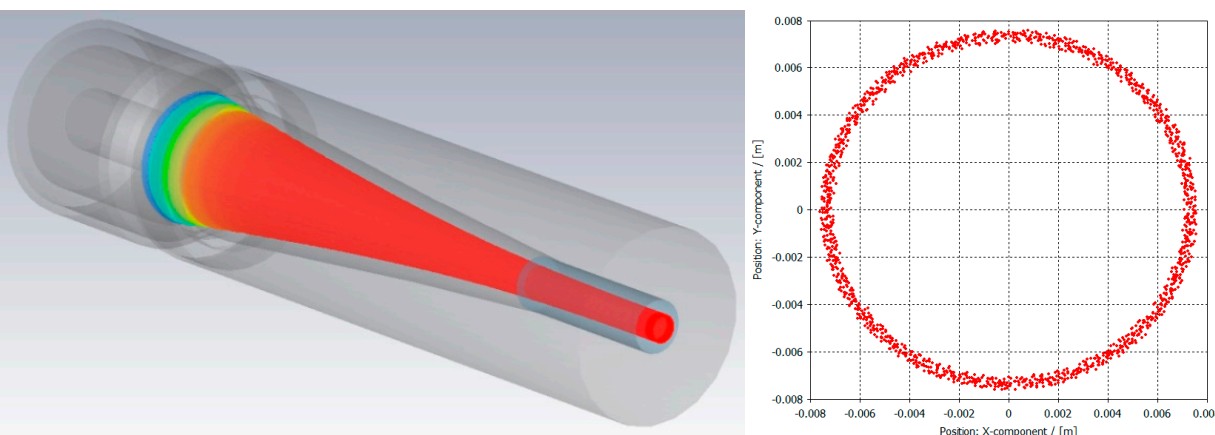

**Figure 7.** Trajectory of the electron beam simulated by CST. In the right image, a thickness of 0.4 mm can be observed for the electron beam injected into the resonator.

In order to provide 1050 °C as the operating temperature of the cathode, we optimized the structure of the MIG and increased the heating efficiency of the fuse, aiming at an idealistic assembly precision. The temperature distribution of the whole MIG was simulated by thermal simulation software, and the nominal voltage and current of the fuse were determined. On the basis of the simulation result, we fabricated an MIG and tested it as close to operating status as possible to validate the temperature while the voltage and current reached nominal values. Figure 8 shows that the experimental temperature of the MIG was about 1048 °C, in contrast with about 1100 °C obtained in the simulation. The reason for the temperature difference is that the ideal connection was used in the simulation, but the thermal resistance between the actual parts was greater.

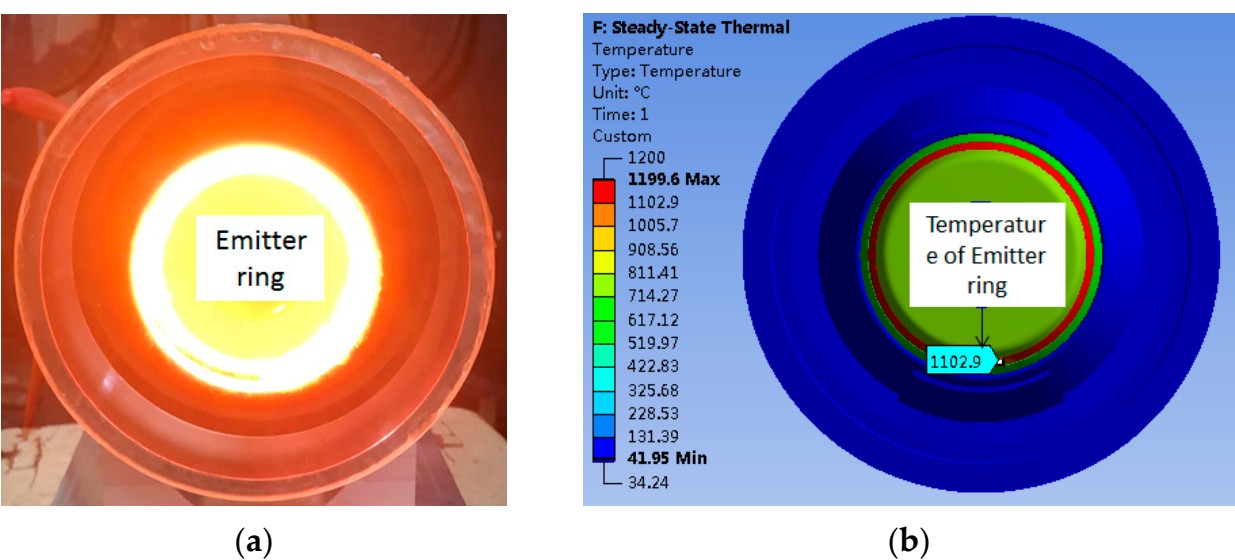

**Figure 8.** Temperature experience and simulation: (**a**) the temperature of the emitter ring (highlight circle) was approximately 1048 °C during testing; (**b**) the temperature of the emitter ring was approximately 1100 °C according to the simulation.

### 3.3. Quasi-Optical Mode Converter

In general, high-power gyrotrons for fusion applications operate with a high-order gallery mode whose mode index is relatively high. Due to the polarization losses and diffraction in the system, such gyrotron resonator modes may transform into the fundamental Gaussian beam, which is suitable for microwave transmission in free space with high efficiency. A quasi-optical mode converter separates the high-frequency radiation

and the collector electron beam, transforms the complex resonator mode into the Gaussian beam with an optimized structure, and minimizes the harmful effect of the high-frequency radiation that may be reflected back into the gyrotron.

The converter consists of a cylindrical waveguide launcher, quasi-ellipse, quasi-parabolic mirrors, and a phase-shaped mirror, which guides the output radiation through a window. For the $TE_{25,10}$ gyrotron, it is necessary to use the Denisov launcher, which has a dimpled-wall surface, to transform the high-order volume mode into a nearly Gaussian beam, as well as to radiate to the mirrors with high efficiency. In the Denisov launcher, the wall perturbations increase the Gaussian mode content compared to the smooth one. Furthermore, the diffraction losses are reduced by decreasing the microwave field amplitude at the cutting area [16,17].

Figure 9 shows the model of the $TE_{25,10}$ Denisov-type launcher, where the radius of the helical converter can be described by the following wall perturbation [18]:

$$R(\theta) = R + \delta_1 \cos(\Delta\beta_1 z - l_1\theta) + \delta_2 \cos(\Delta\beta_2 z - l_2\theta), \tag{4}$$

$$\Delta\beta_1 \approx \frac{1}{2}(\beta_{m-1,n} - \beta_{m+1,n}) = \frac{1}{2}\left[\sqrt{k_0^2 - \left(\frac{\chi_{m-1,n}}{R}\right)^2} - \sqrt{k_0^2 - \left(\frac{\chi_{m+1,n}}{R}\right)^2}\right], \tag{5}$$

$$\Delta\beta_2 \approx \frac{1}{2}(\beta_{m-\Delta m,n+\Delta n} - \beta_{m+\Delta m,n-\Delta n})$$
$$= \frac{1}{2}\left[\sqrt{k_0^2 - \left(\frac{\chi_{m-\Delta m,n+\Delta n}}{R}\right)^2} - \sqrt{k_0^2 - \left(\frac{\chi_{m+\Delta m,n-\Delta n}}{R}\right)^2}\right]. \tag{6}$$

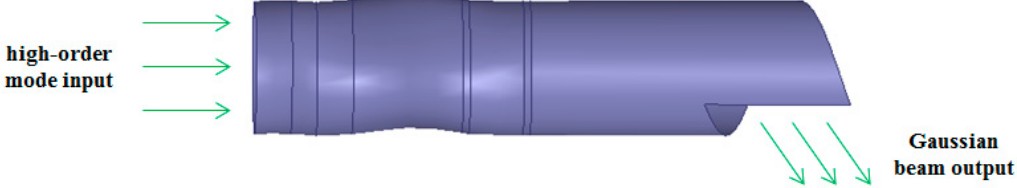

**Figure 9.** Schematic drawing of $TE_{25,10}$ Denisov-type launcher (the radius is increased by 20 times).

In Equations (4)–(6), $l_1 = 1$ and $l_2 = 3$, z is the axial coordinate, and $\theta$ is the angular coordinate; $\delta_1 = 0.04$ mm and $\delta_2 = 0.06$ mm are the amplitudes of launcher deformation. According to the numerical simulations, the efficiency of the launcher was about 99.6%, and the aperture field Gaussian content was approximately 91%. The field distribution of the Denisov launcher is plotted in Figure 10.

In general, some high-order modes were mixed in the fields reflected by the quasi-parabolic mirror and the quasi- elliptical mirror, which propagated to the transmission line at relatively large angles; thus, power was lost in the transmission. In the quasi-optical system, some points do not exist along this direction, because their paths differ in the transmission line. Therefore, phase-correcting methods were used to optimize the distribution and suppress the undesired modes. As shown in Figure 11, the mirror system constituting a phase-correcting mirror was established, where the content of the Gaussian component in the gyrotron window was 92.8%, and the radius of the Gaussian beam was about 50 mm.

To verify the design of the quasi-optical mode converter, we fabricated a measurement system. The $TE_{10}$ mode from the vector network analyzer was converted into the plane wave by an antenna and a phase correcting mirror, and then reflected by a quasi-parabolic mirror injected into the coaxial cavity, which transformed the plane wave into the $TE_{25,10}$ mode as the source of the mode converter measurement system. The output signal was received by another probe of the vector network analyzer, and the amplitude and phase of the aperture field distribution were calculated.

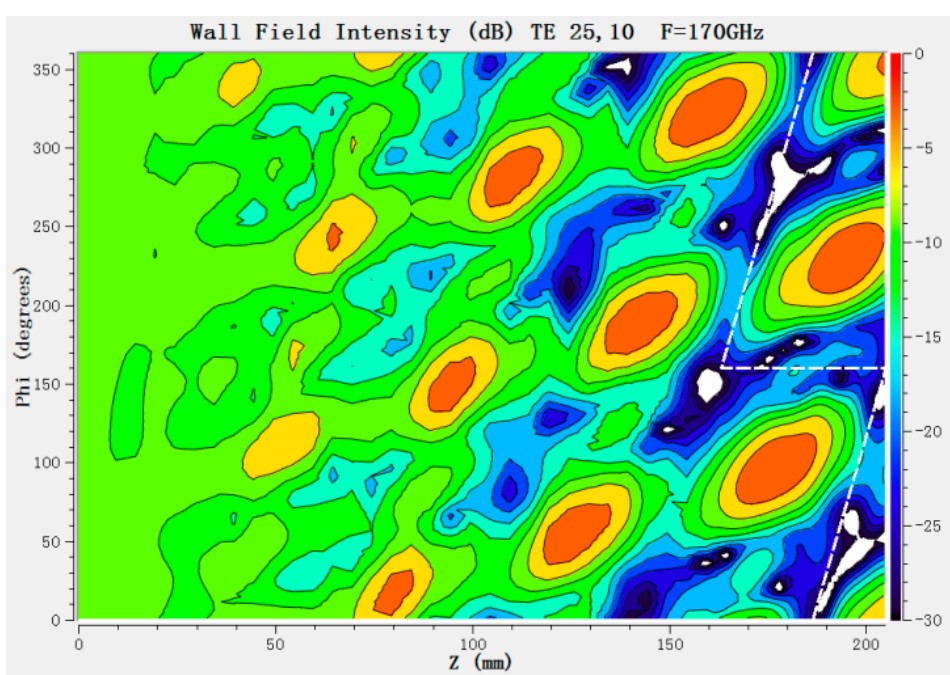

**Figure 10.** Field distribution on the unfolded launcher wall. The white dotted lines denote the edges of the cuts.

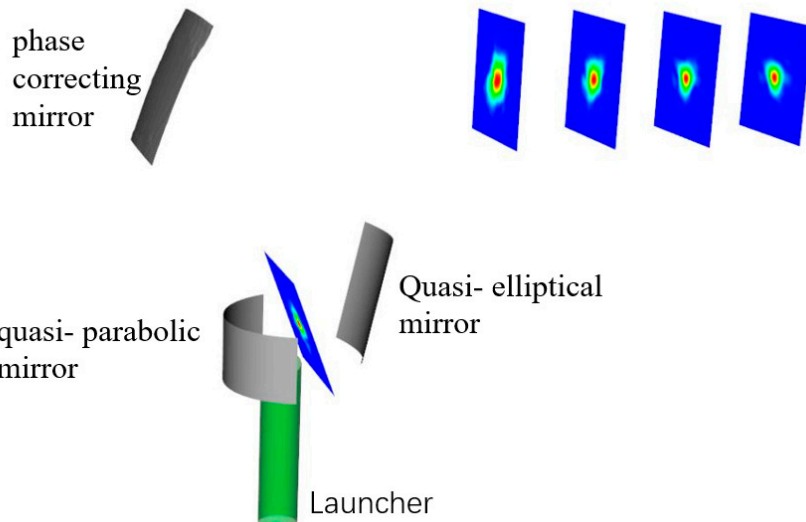

**Figure 11.** Arrangement of the quasi-optical mode converter for the 170 GHz, TE$_{25,10}$ mode gyrotron.

Combined with the BUAA (Beijing University of Aeronautics and Astronautics), we established the TE$_{25,10}$ mode generator and measured the output aperture field distribution, as shown in Figure 12. The experimental results showed that the amplitude and phase were close to the simulated result, but the content of the TE$_{25,10}$ mode component was not ideal. The efficiency of the coaxial cavity was relatively low, while the power reflected by the quasi-parabolic mirror was about −25 dBm, and the output power of mode generator was in the range of −140 dBm to −65 dBm. We are currently optimizing the mode generator and test methods to improve the precision of the mode converter measurement system.

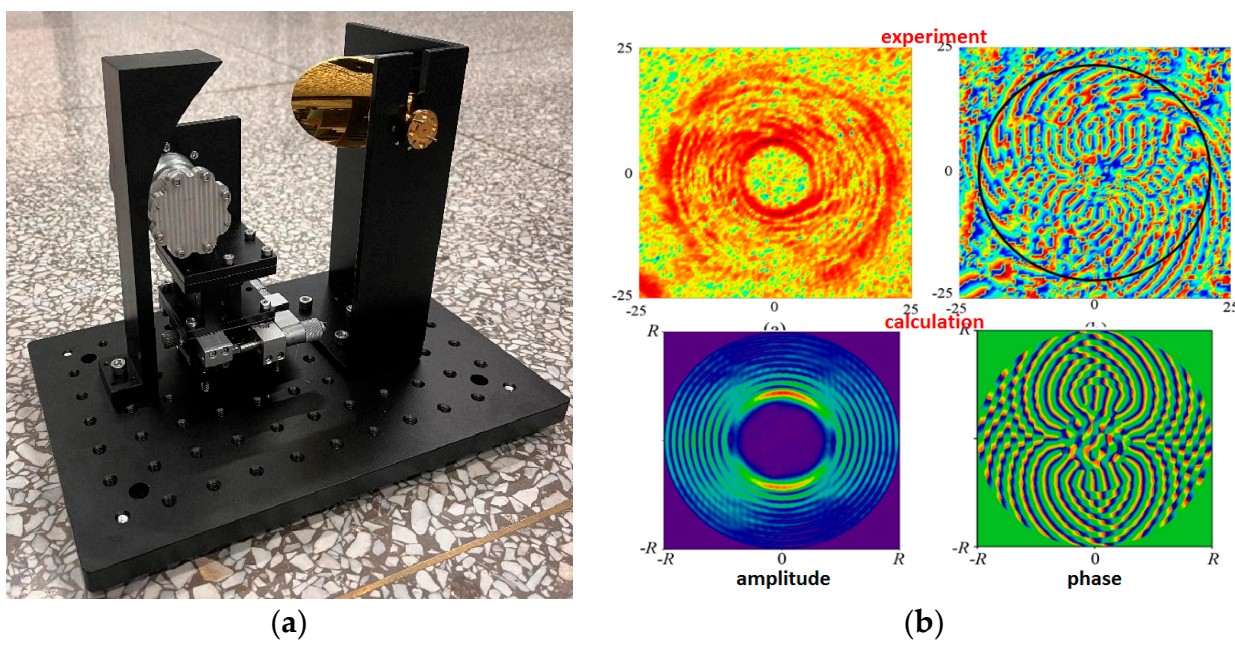

**Figure 12.** TE$_{25,10}$ mode generator: (**a**) established system including a phase correcting mirror, a quasi-parabolic mirror, and a coaxial cavity; (**b**) contrast between the experimental and calculation results for the aperture field distribution of the mode generator output.

### 3.4. Output Window

A single-disc sapphire window was used for the design of short pulse gyrotron. The window was designed for a Gaussian beam, with a window aperture radius of 50 mm, which is twice that of the Gaussian beam radius. The permittivity of the sapphire disc $\varepsilon'_r$ was 9.4, and its thickness was 1.4 mm.

Figure 13a shows the measurement system we used to verify the transmission performance of the window disc, where microwave beam emitted from a vector network analyzer is transformed into the Gaussian beam by the antenna and mirrors, so that the S parameters of the window disc in the fundamental Gaussian beam can be measured. The contrast between experiment and calculation is shown in Figure 13b, showing that the power loss of the window was less than 1%.

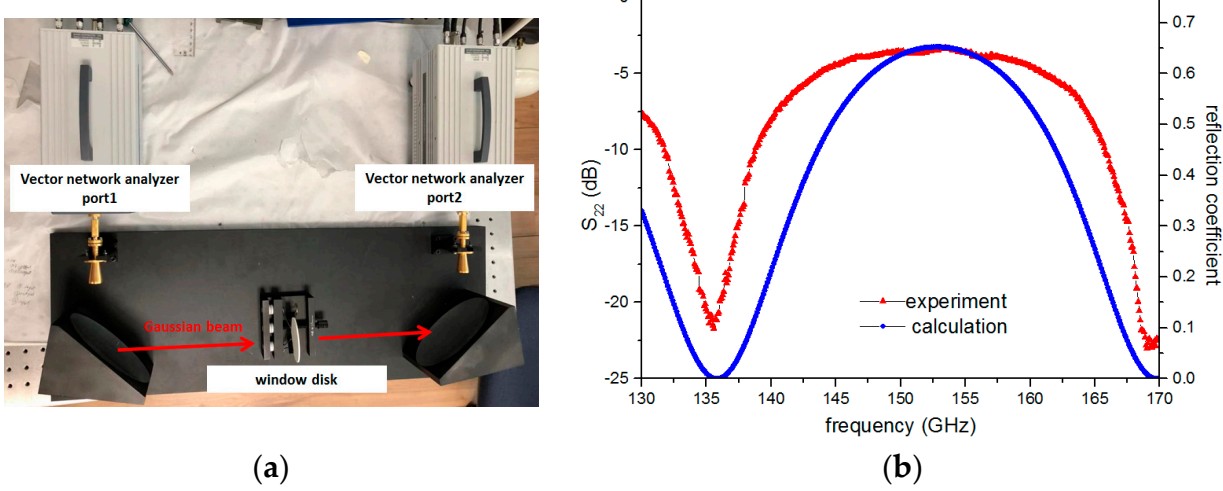

**Figure 13.** Measurement system and result of window measurement: (**a**) measurement system of window; (**b**) experimentally measured and calculated reflection coefficients of sapphire window.

## 4. Exploration Testing

For the experimental testing of the 170 GHz gyrotron, the magnetic guidance system and high-voltage power supply system were upgraded. A superconducting magnet with a maximum magnetic field of 7 T was used, and the distribution of the axial magnetic field was optimized to adapt to the electron beam of the 170 GHz gyrotron. The output power of the supply system was upgraded to 400 kVA, and the maximum value of voltage could reach 80 kV, satisfying the gyrotron operation for a pulse length of 100 ms. The superconducting magnet and power supply are shown in Figure 14.

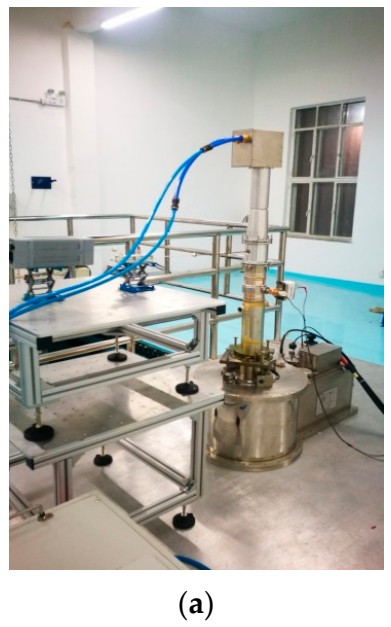
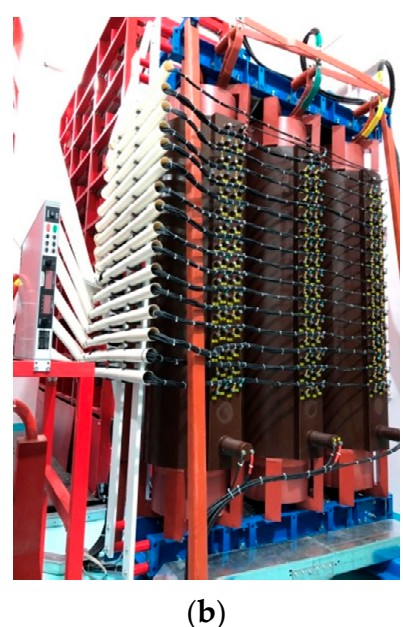

(**a**)　　　　　　　　　　　　　　　　　　　　　　　　　(**b**)

**Figure 14.** The 170 GHz gyrotron measurement system consisting of a superconducting magnet (**a**) and a high-voltage power supply (**b**).

On the basis of the simulated result, a 170 GHz gyrotron without a quasi-optical mode converter was fabricated to validate the electronic–optic system and RF system. It is currently under testing, with a peak power of about 210 kW for an accelerating voltage of 60 kV and current of 25 A (the monitoring waveform is shown in Figure 15). Its efficiency was 15.9% at a pulse length of 0.1 ms. During the experiment, the output power was sensitive to the magnetic field; thus, it was necessary to confirm the magnetic field in a precise range. The $TE_{25,10}$ mode could operate steadily, and the pure frequency spectrum could be observed. Due to the voltage and current being below the nominal value, the output power and efficiency was only 210 kW. Under the experimental voltage and current (60 kV and 25 A), the simulation result of the cavity showed that the output power of the cavity was approximately 250 kW, which is close to the actual test value considering the influence of transmission loss and spread of electron velocity. The accelerating voltage and current to reach the nominal value can be continuously improved by tuning the modulating anode voltage and magnetic field.

A gyrotron with a quasi-optical mode converter will be fabricated as soon as possible to validate the quasi-optical mode converter performance and the heat dissipation design of all parts. At the next step, a higher resonator mode will be considered to further improve the power generation of the 170 GHz gyrotron. The resonator diameter can be increased by increasing the mode number, which will reduce the heat load density of resonator wall. The effective heat management of long-pulse power modulation for fusion applications is also under study.

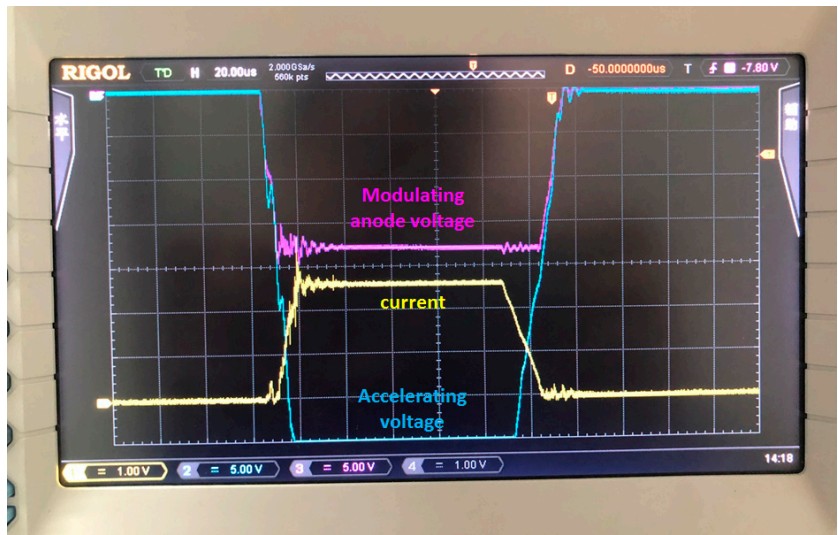

**Figure 15.** Measured voltage (modulating anode and main anode) and current of the 170 GHz gyrotron.

## 5. Conclusions

The feasibility and conceptual design studies of a 170 GHz, $TE_{25,10}$ mode gyrotron were presented for ECRH applications in thermonuclear fusion reactors. A gyrotron with 33.1% theoretical efficiency was designed at the power level of 1 MW using a conventional hollow cavity. By calculating the mode competition and starting current, the $TE_{25,10}$ mode was selected as the gyrotron operating mode, yielding satisfactory results. A triode-type MIG was designed and optimized to decrease the spread of velocity. The mode converter consisting of a dimpled launcher and three mirrors ensured that the content of Gaussian component in the gyrotron window was 92.8%. Cold test measurements of the mode converter and output window were performed, revealing approximately 1% power loss of the output window, and the mode generator was optimized. In addition, a prototype gyrotron used to validate electronic–optic system and RF system was fabricated, whose peak power was approximately 210 kW for a short pulse.

**Author Contributions:** Conceptualization, Y.Z., X.Z. and J.F.; methodology, M.B., M.J. and Q.L.; validation, Y.Z.; formal analysis, Y.Z. and W.H.; investigation, Y.Z. and X.Z.; resources, J.F.; data curation, Y.Z.; writing—original draft preparation, Y.Z.; writing—review and editing, Y.Z., X.Z. and J.F.; visualization, D.G.; supervision, J.F.; project administration, Y.Z.; funding acquisition, Y.Z. All authors have read and agreed to the published version of the manuscript.

**Funding:** This work supported by the National magnetic Confinement Nuclear Fusion Power Special Fund of China (Grant No. 2017YFE0300203).

**Data Availability Statement:** All data included in this study are available upon request from the corresponding author.

**Conflicts of Interest:** The authors declare no conflict of interest.

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
