# Peer review of "The Development of 170 GHz, 1 MW Gyrotron for Fusion Application"

_electronics, doi:10.3390/electronics11081279_

Round 1

Reviewer 1 Report

This is an interesting and well-constructed paper on the design of a 170 GHz gyrotron for fusion application. I believe this paper is worthy of publication, subject to a small number of clarifications and additions:

  • Abstract, line 19: here, the authors describe the system having an output power of 210 kW and efficiency of 15.9%. On reading the rest of the paper, this is clear that it is a low power test; however, this could be made clearer in the abstract.
  • Introduction - additional references would be welcome, e.g. historical papers for gyrotron (e.g. Gaponov 1977; review paper Chu 2004).
  • Line 66 - change the subscript of TEm,n mode.
  • Table 2, line 75 - Some of the variables are unclear as to what they are for each mode, e.g. column 3, 4, and 5.
  • Line 98, figure 3 caption: "Normalized" instead of "Nominalized"?
  • Line 102, "taper" instead of "tapper".
  • Table 3: Here, voltage, U_C, is used. Previously accelerating voltage is used, and on the next page, Uc, is used. It would be good to be consistent.
  • Line 156 - how was the MIG structure optimised? With CST?
  • Line 239, & Figure 13 - Why the large discrepancy in bandwidth between experiment and calculation? Is this due to the manufacturing of the window, or the measurement setup?

Author Response

Thanks for your response. I have modified all sugestions for the manuscript and marked up using the “Track Changes” function. In addition, I have read the manuscript carefully and modified some contents to express my own points clearly, and the modified contents does not involve calculation and theoretical analysis results.

Reviewer 2 Report

The manuscript presents interesting results of design, development, and testing of a 1-MW gyrotron for fusion. The content is appropriate for the journal and seems interesting and important for the community. However, the clarity of presentation should be improved. Main critical points are listed below.

  1. The manuscript definitely needs a careful language editing. In particular, there are many typos and grammar mistakes that should be corrected.
  2. Page 2, line 45: "the electron gun" should be replaced with "The electron gun".
  3. Page 2, line 51: "the electron beam interacts with the millimeter-wave" should be replaced with "the electron beam interacts with the millimeter-wave radiation."
  4. Page 2, line 57: "had completed the all design" should be replaced with "had completed all design" or "had completed the design."
  5. Page 3, line 67: "is the nth root of ??′ (?)=0" should be replaced with "?mn is the nth root of ??′ (?)=0". "the radius" should be replaced with "The radius".
  6. Page 3, eq. 2. Please mention that in ??±1,1, "±" corresponds to co-rotating and counter-rotating mode.
  7. Page 3. Please clarify what kind of theory was used to calculate the start-oscillation currents presented in Figure 2.  If the calculations are presented for the gyrotron cavity parameters listed in Table 3, please clarify. What is shown with thin lines in Fig. 2? Why there are two nearly identical curves (red and green) for the TE23,10 mode?
  8. Page 3, line 88: "TE25,10,1" should be replaced with "TE25,10,1"
  9. Page 4, line 102: "tapper" should be replaced with "taper".
  10. Page 5, lines 109-110: An awkward phrase "Figure 5 shows the function of output power and magnetic field in the TE25,10 mode, and the beam parameters corresponding to CW operation at the current of 40A, and the accelerating voltage of 80kV." Please reformulate as "Figure 5 shows the output power versus magnetic field for the TE25,10 mode with beam current of 40A and the accelerating voltage of 80kV, which correspond to CW operation.
  11. Page 5, line 113: "for 170 GHz resonator, the parameters are shown in Table 3" should be replaced with for 170 GHz resonator with the parameters shown in Table 3." 
  12. Same line: "power is 1.06MW". However, according to Fig. 5 it seems that for the TE25,10 mode the maximal power is ~ 1.1 MW.
  13. Fig. 5, caption: "Simulation result of cavity" should be replaced with "Simulation result of the gyrotron with cavity parameters listed in table 3", or in some other way.
  14. Page 6, lines 125-126. Please reformulate the sentence "To further attenuate possible RF fields, a new beam tunnel structure consisting of several copper rings and atten-126 uating ceramic rings (BeO) was used ... " It is not clear whether you are speaking on the gyrotron presented in [12] or on your own design.
  15. Page 6, line 144: "approximate 0.29T" should be replaced with "approximately 0.29T".
  16. Page 6, lines 145-146: "The numerical calculations have been done using 2D code and CST trajectory studio". Please explain what 2D code was actually used in the simulation. Please comment on the agreement between this code and CST?
  17. Caption to Fig. 8: "Picture of MIG". Well, it seems that in Fig. 8(a) you presnet a foto of the gun, while on Fig 8(b) there is a result of numerical modeling. Please clarify.
  18.  Page 10, line 223. Please spell out the acronym BUAA. Please correct "we have been established" to "we established".
  19. Page 10, line 224: Do you really mean figure 10 or figure 12?
  20. Page 10, line 236: "Figure 11 (a) is a measurement system" should be replaced with "Figure 11 (a) (probably, Fig. 13(a)?) presents a foto of the measurement system."
  21. Caption to Fig. 13:  "result of window" should be replaced with "results of window measurement",  "experimental and calculation result of sapphire window" should be replaced with "experimentally meadured and calculated reflection coefficшуте of the sapphire window". 
  22. Page 10, line 250: "The output power of the supply system upgraded" should be replaced with "The output power of the supply system is upgraded."
  23. Page 11, lines 262-265. The sentence "the voltage and current of experiment value (60 kV and 25A) were used to simulate the cavity, the output power of cavity is approximately 250kW, which is close to the actual test value considering the influence of transmission loss and spread of electron velocity" is very poorly formulated. Please revise.  
  24. Page 11, line 266: "modulate anode voltage" should be replaced with "modulating anode voltage."
  25. Page 11, line 270: I can't understand what is "quasi-mode output gyrotron". Do you mean the gyrotron with a quasi-optical mode converter?
  26. Page 11, lines 274-275: "And feasibility of long pulse power modulation for fusion applications is also studied." Did you really study the long-pulse operation? Or "will be studied"?
  27. Page 12, line 279: "Theoretical efficiency of 33.1% has been designed." You can design a gyrotron with 33.1% efficiency, but not the efficiency.
  28. Page 12, line 285: "Cold test methods have been developed for mode converter and output window."   Maybe "Cold-test measurements od the mode converter and output window have been performed"?
  29. Ref. 2: missing journal title. 

To summarize, please read the manuscript carefully and make the neccessary revisions.

Author Response

(The authors gave the same response as above.)
